# Futurism without a Future: Thoughts on *The Ministry of Time* and *Mirage* (2015–2018)

Victor M. Pueyo Zoco

Department of Spanish and Portuguese, Temple University, Philadelphia, PA 19122, USA;
victor.pueyo.zoco@temple.edu

**Abstract:** The future is not what it used to be. A new strain of futurism has taken over the stage of global science-fiction: one whose understanding of the future cannot be distinguished from its understanding of the present. Gone are the days when extraterrestrials in shiny, extravagant outfits mastered fascinating technologies that flirted with magic. Characters in Charlie Brooker's *Black Mirror* (2015–2020) dress like us, and the dystopian technology they put up with is, for the most part, a technology that has existed for years. Armando Iannucci's imagining of a space cruise for rich people in *Avenue 5* (2020) overlaps with Elon Musk's actual plans of sending wealthy tourists to the moon, while Albert Robida's visionary *téléphonoscope* (1879) amounts to a sad reminder of our everyday Zoom call. Is not the current COVID-19 crisis the blueprint to the ultimate post-apocalyptic script? Spanish filmmaker Juan Antonio Bayona noted in a recent interview that Steve Soderbergh's *Contagion* (2011), originally labeled as a sci-fi movie by IMDB, is now a *drama* according to the same internet portal. Science is not fiction anymore, which means at least two different things: that science has lost the power to convey the kind of awe that may be later turned into fiction, and that fiction seems to be unable to inspire a narrative of scientific or—broadly speaking—human progress. How can we retrieve the emancipatory value of progress in good old futuristic sci-fi when the future coincides with the present? What *should* cultural production look like to help us imagine an alternative to financial capitalism in the face of the impossibility of utopia? The answer, I will claim, resides in Franco Berardi's concept of "futurability". This paper explores the limits of this concept by reading side by side Javier Olivares' and Pablo Olivares' *The Ministry of Time* (2015) and Oriol Paulo's *Mirage* (2018).

**Keywords:** capitalist realism; futurability; prototopia; post-fordism; immanence; Spanish cinema; *El ministerio del tiempo* (*The Ministry of Time*); *Durante la tormenta* (*Mirage*)

## 1. Introduction

> "*Fin du monde, fin du mois, même coupable, même combat*" CGT slogan, 1 May 2019 (Lyon, France)

We will begin with a case of unintentional comedy, included in André Breton's *Anthology of Black Humor* (1940). Charles Fourier, an enlightened temperament when discussing matters of political economy over dinner with Owen or Saint-Simon, also exhibited a tendency to indulge in terrible digressions and to endorse the most delirious visions of future worlds. The *Treatise of the Domestic-Agricultural Association* is just another example of his rare penchant for the impossible. Fourier's faith in science led him to believe that it would be possible, at some point, to reverse the scene of creation through an effort of "countermoulding", by which "he who gave us the lion will give us as countermold a superb and docile quadruped, an elastic carrier, the anti-lion, the kind of post-animal that would allow a rider, who leaves Calais or Brussels in the morning, to lunch in Paris, dine in Lyons, and sup in Marseilles" (Breton 1997, p. 47). Horses, Fourier points out, are rude and simple creatures that will be employed for light harnessing and fancy parades

once we possess a whole family of elastic steeds. The anti-lion, the anti-tiger, and the anti-leopard will easily cover eight yards with each bound, while the rider, on the back of their charger, will be as comfortably installed as if in a well-suspended berlin, and that is only the beginning. Fourier believes that a new breed of negative beasts will transform—within *five* years—every domain of terrestrial and marine life: there will be anti-crocodiles working as gondoliers, anti-sharks helping track down the fish, anti-whales towing vessels through the calm waters of the sea, etc.

Now, even though these fantastic animals already populate our late-capitalist social formations, whether we call them automobiles, tow trucks, trains, or radars, Fourier's futuristic fantasy is far from being a story of prophetic success. What is so irresistibly funny about it? Perhaps, I would argue, that Fourier imagines otherness as simply negating the same, yet retaining its most characteristic traits. We try to imagine the anti-shark as a sophisticated fishing device, but that device remains a shark, looks like a shark, and swims like a shark. We try to imagine the anti-tiger as an elastic horse (whatever that might mean), but we never stop visualizing the ridiculous picture of a tiger pulling a stagecoach or a barouche. A famous quote by Groucho Marx comes to mind: "He may look like an idiot and talk like an idiot but don't let that fool you. He really is an idiot". Appearances cannot be so easily overlooked because they are not just appearances; they are matter, and matter, whether theory likes it or not, *matters*.

Fourier's counterfactual thinking shows, if anything, the poverty of dialectics, or how dialectics can only go so far. Pure negativity is the real fantasy here: to envision the other as a negation is to imagine the same otherwise.[1] In other words: the lesson to be learned from Fourier's domestic-agricultural association is that progress may end up taking place not thanks to our ability to imagine it, but despite it. Reading Fredric Jameson or Mark Fisher, who famously claimed that "it is easier to imagine the end of the world than to imagine the end of capitalism" (Fisher 2009, pp. 1–12), encourages us to long for an Edenic world of symbolic prosperity, a 'purely modern' moment in the development of capitalist relations of production wherein it was not only possible but even customary to imagine a broad range of possible futures.

We need to be ready to admit that this moment may have never existed. Early modern utopian texts such as Thomas Moore's *Utopia* or Tommasso Campanella's *The City of the Sun* depict inevitably early modern societies, whereas their enlightened counterparts, for example Voltaire's *Candide* or the anonymous Spanish *Sinapia*, portray an alternative world carefully fashioned after enlightened principles. Just as Fourier's anti-tigers are the opposite of tigers, Sinapia is the *antipode* country of Hispania or Spain. Similarly, Samuel Butler's *Erewhon* could only make sense because the title of the novel is Nowhere read backward: the opposite of something that is opposing itself. Is this not Karl Marx's *Das Kapital* Hegelian trademark and its main flaw? Could not we say, in hindsight, that its best achievement was not to give birth to the idea of communism per se, but to deliver the possibility of an anti-capitalist ethos? The radical relativity of utopian thinking makes the question at stake even more relevant and appealing: if utopias have always relied on the reproduction of an already given *topos* to be meaningful, and if modernity was never able to transcend itself the way we sometimes assume it did, how are we supposed to overcome our own symbolic conditions of existence and imagine the future otherwise? How can the future be thought from immanence? Late capitalism, I claim, provides an invaluable vantage point to answer this question. Fisher coined the term "capitalist realism" to signal a current state of things according to which it is becoming harder and harder to envisage the outside of a late capitalist framework that is as all-encompassing and pervasive as any ideology has ever been. Capitalist realism would designate "the widespread sense that not only is capitalism the only viable political and economic system, but also that it is now impossible even to imagine a coherent alternative to it" (Fisher 2009, p. 6).

In the 1980s, when Jameson first advanced his thesis about postmodernism, there were still, in name at least, political alternatives to capitalism [ . . . ] A whole generation has passed since the collapse of the Berlin Wall. In the 1960s and 1970s, capitalism had to face

the problem of how to contain and absorb energies from outside. It now, in fact, has the opposite problem; having all-too successfully incorporated externality, how can it function without an outside it can colonize and appropriate? (Fisher 2009, pp. 7–8).

Fisher's diagnosis is most assuredly right, but also, to some extent, an example of the "capitalist-realist" outlook he decries in his book. My take on this is a little more optimistic. It is precisely because there are no alternatives (or because alternatives have been narrowed down to a bare minimum today) that we are faced with the necessary challenge of thinking the future non-dialectically. In so doing, we can finally escape the cul-de-sac of idealistic leftist utopias that are either devoid of real content (presenting themselves as "ideas", as naked negativity) or surreptitiously equipped with the substance of what they seem to deny. In both cases, the dialectic gesture, implicit or not, privileges an absolute otherness that posits itself as difference when it is nothing but repetition. In what follows, I explain how contemporary science-fiction, or a certain breed of science-fiction that I identify with Fisher's notion of capitalist realism, can help us assess the different futures that are already unfolding before our eyes. I focus on two Spanish science fiction works: *El Ministerio del tiempo* by Javier Olivares and Pablo Olivares and *Durante la tormenta* by Oriol Paulo. Deeply rooted in the muddy waters of capitalist realism, these works represent contradictory tendencies, even mutually exclusive approaches to the question of how to apprehend that elusive thing we used to call the future (elusive, partly, because it is too much within reach). This is, of course, an old debate. The future has been contemporary to us since Francis Fukuyama diagnosed "the end of history", but it is only now, when there is nothing else to look at, that we can see it for what it is: an orphaned ghost, a moving sequence, an empty chapter in the book of the present.[2]

## 2. Future Pasts, or How We (Used to) Remember the Present

No matter how hard or unfeasible it was to imagine different futures, no matter how illusory or deceitful, it is true that the future used to be a thing. Looking back at the many old-timey futuristic fictions filmed during the 1960s and the 1970s (from *Star Trek* and *Lost in Space* to the original *Battlestar Galactica* and *Buck Rogers*), one cannot but note that futurism looks already retro today. It is not so much that the technology featured in these shows is largely outdated—teleportation and interstellar trips, unless I missed the last memo, still belong to the realm of wonder. When I say that the future looks retro today, what I mean is that the very idea of imagining the future, of translating the future into images, has a vintage feel to it. This has little to do with the content of what we imagine. The good causes that were fought in deep space back then (racial equity, social justice, ecological sustainability, or stopping imperial forces from harvesting "the spice" in underdeveloped planets) are still the causes fought today. The problem lies, rather, in the idea of imagining the future itself, which is what we regard as an anachronism. There is nothing inherently retro about tinfoil outfits, methacrylate objects, waterdrop-shaped cars, or bicolored onesies, as there was nothing inherently futuristic about them in the first place; the only thing that is retro about these common sci-fi places is the fact that they were meant to signify that the future was different from the present. It is this semiosis of estrangement or *Verfremdung* that is outdated, rather than the elements of which it is comprised.[3] "The wonderful future that never was", as Gregory Benford (2010) would put it, has yielded to "the horrendous present that never stops being", which means, in practice, that, to a large extent, we cannot distinguish the present from the future anymore.[4]

Contemporary futuristic fantasies resemble our everyday pedestrian life in such a detailed fashion that we sometimes forget that the events we are witnessing are set in the future. More often than not, science fiction has been replaced by a technological drama of sorts, whereby the dystopian events narrated merely augment the social effects of a technology that already exists. Charlie Brooker's *Black Mirror* (Brooker 2011) is a notorious example of this trend, as is Koldo Serra's *Distopía* (Serra 2014), the Spanish response to *Black Mirror*. In *Distopía*'s pilot, "Ciudadanos", two citizens kidnap the minister of economy to let people decide through an internet app whether he should live or die. Set against the

backdrop of the 15-M protests in 2011, the series succeeds, if anything, at translating the impotence of the kidnappers into narrative impotence. There is no trace of any dystopian element that might justify the choice of its name; in this brand of fiction, reality is not dystopian; dystopia, if anything, is real. Meanwhile, in *Black Mirror*'s opening episode of the third season, "Nosedive", Lacie is a young woman obsessed with improving her personal rating in a world where people can rate each other from one to five stars for every interaction they sustain. The rating is crucial because it impacts their socioeconomic status. Lacie needs to raise her rating from 4.2 to 4.5 to afford her dream luxury condo. Her desperate attempts to please everyone around her will lead to disastrous consequences, as she sees her rating reduced to less than 1 point and ends up in jail. Needless to say, this plot would have made for a nice dystopian fantasy in the 1970s, one starring the likes of Charlton Heston or Michael York (think of *Soylent Green* or *Logan's Run*); now, amidst the symbolic storm of capitalist realism, the episode hardly qualifies as dystopian. The mobile app *Peeple*, developed by Nicole McCullough and Julia Cordray, allows people to write recommendations for others based on professional, personal, and romantic relationships. One could argue that the cannibalistic dystopia of *Soylent Green* ("Soylent green is people!") has finally come full circle. If *Peeple* is frequently celebrated as "the Yelp of people", that can only mean one thing: that we are the food now.

The question here is the easiest one: how is the very existence of apps such as *Peeple* substantially different from the science fiction narrative with which we are presented in "Nosedive"? Moreover, how do *Peeple* and "Nosedive" differ, in essence, from the national credit rating system that is being developed by the government of the People's Republic of China, which is basically a tool to facilitate white and blacklisting? Is China experiencing a shift from populism to *Peeple*ism? Let us not forget, in passing, that China's leading mass surveillance system is called Skynet, a name borrowed from dystopian sci-fi franchise *Terminator*. China proves once again, as Slavoj Žižek has repeatedly warned, that there is no natural correlation between capitalism and liberal democracy: capitalism does very well under totalitarian conditions when it does not clear the way for them (Badiou and Žižek 2012, pp. 103–7). However, we should not need to go as far away as to the Far East or the end of the world to make a simple point. Most of the population in the United States consistently ignore that most countries do not have a credit score system, whereas most of the population in other countries ignore that there is such thing as a credit score system in the United States. Outside of the United States, the credit score system is usually compared—can only be compared—to *Black Mirror*. Perhaps that is why the mirror is imagined as black or opaque: fiction does not mirror reality anymore; it is just reality *imitating itself*.[5]

The explanation for these similarities is an unsettling one. Let us say, in plain words, that the distance that separates the real from the possible, or history from poetry, in Aristotelian terms, has shrunk dramatically over the last three decades. This is no petty statement. The obsession with verisimilitude (the name we used to give to such distance) can be traced back to the birth of modern literary thought in the sixteenth century, along with other terms such as *mimesis* or *enargeia* (Matamoro 1987, pp. 83–101). For Robortelli, Minturno, or Scaligero in Italy (or El Brocense, López Pinciano, and Cascales in Spain), verisimilitude was, to a large extent, a tacit agreement as to what could be said and what could not, which is why some authors analyze the concept of verisimilitude as a juridical concept. As Jesús Rodríguez Velasco notes: "The task of the law consists in ensuring that the words and stories, whether or not they correspond to a reality that is otherwise incomprehensible, remain inside the limits of verisimilitude" (Rodríguez Velasco 2006, p. 70). First and foremost, truth had to *ad-just* (etymologically, "walk towards fairness") to the rule of the possible. Since the discussion of the juridical origins of verisimilitude would sprout a long and productive theoretical debate that exceeds the scope of this paper, let us just state the obvious for now: de-regulation—economic, symbolic, and otherwise—has made this distinction between the real and the possible obsolete within the framework of the financial capitalist world-system. The real and the possible now sit one alongside

the other according to a forcible horizontal distribution of the sensible, to partially misuse Jacques Rancière's expression (Rancière 1999), that seems unable to tell them apart. As a result of that, Armando Iannucci's imagining of a space cruise for billionaires in *Avenue 5* (2020) overlaps with Elon Musk's actual plans of sending wealthy tourists to colonize the moon, while Albert Robida's visionary *téléphonoscope*, sketched out in his wonderful novel *Le Vingtième Siècle* (1879), amounts today to a fastidious reminder of our everyday Zoom call. Finally, and speaking of Zoom calls, many have remarked that the current COVID-19 crisis acts in practice as the blueprint to the ultimate post-apocalyptic script. Acclaimed filmmaker Juan Antonio Bayona recalled in a recent interview with journalist Jordi Évole (Évole 2020) that Steve Soderbergh's *Contagion* (2011), originally labeled as a sci-fi movie by IMDB, is now cataloged as a drama according to the same internet portal (2020, 1:50). Similarly, *The Martian* won the Golden Globe award for Best Comedy in 2016. Even the new *Matrix* sequel becomes meta to explore this premise, which calls into question, once more, if there is any other premise left to explore.

As the border between the real and the possible blurs, movies and television start dealing with a new brand of unfiltered reality, be that of atrocious reality TV (Kardashians, Snookies, etc.) or that of "found footage" horror movies such as Oren Peli's *Paranormal Activity* (2007), Paco Plaza and Jaume Balagueró's *REC* (2007), and Matt Reeves' *Cloverfield* (2008).[6] Only by assessing this phenomenon correctly can we begin to understand that contemporary science fiction plays, in late capitalist social formations, a role that mimics the role the realist novel played within the coordinates of nineteenth-century classic capitalism. Where the realist novel grasped reality by negotiating the gap between the public and the private (e.g., by representing lives), contemporary science fiction comes to terms with it by enacting its erasure. Since the real and the possible coincide in these movies, science fiction becomes the most accurate tool to capture the state of things that Fisher called capitalist realism, which is, roughly speaking, the world in which we live today. That is why these fictions should not be taken lightly. We need science-fiction more than ever because capitalist realism—frequently wrapped up in sci-fi packaging or camouflaged as unassuming horror cinema—is the new brand of realism today.

I will give one last example. Ron D. Moore, the co-creator of the hugely popular *Battlestar Galactica*'s remake, penned in 2003 a manifesto for a "naturalistic science fiction", the *Battlestar Galactica Series Bible*. It aimed to "introduce realism into what has heretofore been an aggressively unrealistic genre" (Moore 2003, p. 1). To do so, the show welcomed techniques that had not been precisely germane to sci-fi productions in the past: the resort to documentary or cinema verité style (hand-held camera, practical lightning, a functional set design); the avoidance of both MTV's fast-cutting and Star Trek's "master, two-shot, close-up, close-up, two-shot, back to master" characteristic pattern, etc. Moore also embraced an almost Dogma 95 approach to ambiance-building ("Our spaceships don't make noise because there is no noise in space"), along with the rejection of CGI gimmicks and 3D hero shots "panning and zooming wildly with the touch of a mousepad" (Moore 2003, pp. 1–2). His is, without doubt, a laudable effort. However, Moore's somewhat pretentious manifesto has fallen victim to its own success. What was meant to be a groundbreaking program now defines the conventions of almost every sci-fi movie and TV show airing on streaming platforms (zombie soap operas or the above-referenced found footage horror films are among them). Realism was never an "artistic" issue, but an ontological one: how can a naturalist sci-fi manifesto challenge the limits of a reality that is already *blatantly* naturalistic?

Thus, the question as to what kind of science fiction we need remains still unanswered. Is dystopian sci-fi the answer? After all, a world that is already condemned (overrun by zombies or devastated by an ecological catastrophe) confronts us with an ethical disjunctive: either the characters will try to save themselves, neglecting the others, or they will work together for the collective benefit of the group. I will point out something for now: there is, to the best of my surely limited recollection, not a single movie or TV series of those labeled as "post-apocalyptic" that advocates for their selfish characters. All these movies make the

right choice for us; they assume that solidarity is the way to go and they kill the characters that are reluctant to abide by it. Take as an example, if you wish, Juan Carlos Fresnadillo killing off his main character (played by Robert Carlyle) in *28 Weeks Later* (2007). Yet, the ethical stance that favors solidarity over other life options does not seem to bear significant fruit, unless we truly believe the world has improved a lot during the decades in which this narrative outlook has been the predominant one. Is utopian sci-fi the alternative? Maybe, many will say, if instead of exaggerating the negative traits of current social formations, we focused on the depiction of an ideal future society and how it might overcome all the obstacles our political adversaries deem unsurmountable, we would be able to safeguard the good values while clearing a viable path to an unwritten future. If the problem is that we are stuck in the present, it is only natural to suppose, as the explorer in the opening sequence of Mark Dennis' *Time Trap* (2017), that "the answer [to our being trapped in the present] is in the future" (Dennis 2017, 2:15). However, is this mirage of a future haven not the real time trap that locks down the inner possibilities of the present? Furthermore, we are back to the conundrum that Jameson and Fisher exposed years ago: How do we imagine (not represent, but imagine) that of which there is no image available, to begin with? Moreover, how do we get people to bear with us and accept those are the "good values"—values that are radically incompatible with their inherited worldview—without losing the audience to a wide-open yawn before reaching the ten-minute mark?

The answer is, of course, a type of fiction that is neither utopian nor dystopian, but just strategically consistent with the capitalist-realist ideological environment to which all these dystopias (and willy-nilly utopias too) are already native. Reflecting on Michel Foucault's concept of heterotopia, Walter Russell Mead has written: "Utopia is a place where everything is good; dystopia is a place where everything is bad; heterotopia is where things are different—that is, a collection whose members have few or no intelligible connections with one another" (Mead 1995, p. 13). Anyone who is not a total dummy, and I sincerely doubt that whoever is reading this article can qualify as a dummy at all, will notice that Mead's synoptic description of all possible topological relations is lacking the implied fourth element. If dystopia is the negation of the good place utopia (or *eutopia*) is set to represent, there should be a logical correlative for that place where all things are different (heterotopia): the place where all things are the same. I will call this place *prototopia*.

Prototopia is the place that is always already (*proto-*) occupied by itself. Consider, for instance, what a mall is. A mall is different from a department store in that it is not divided into floors (first floor, women's clothes; second floor, toys and male garments; top floor, cafeteria, etc.). A mall is designed as a never-ending streamlined surface where there is no actual separation between the different stores of which it is comprised. Shoes, appliances, restaurants, massage parlors: they all chaotically flow across the space without reclaiming a previously assigned place in the whole. Yet, we are not presented with a disparate collection of elements: all of them are connected by the idea of the mall, which is already making sense of the whole from within, just as the water makes sense of the glass. We could say the stores belong to the mall only inasmuch as the mall belongs to itself. Deleuze and Guattari (2000) would call this a "body without organs", but the mall is not naturally there as a body is.[7] Rather, it has been planned and conceived following a certain pattern: that of an empty surface that takes the place of the truly empty space to become its content (think of an Apple store or those modern university libraries where books are not visible or accessible anymore, displaced by the space that takes over the space). In Spanish, we call malls "grandes superficies" ("large surfaces"), highlighting the fact that the most important thing about them is not their being large, but their being *surfaces*—etymologically, whatever is above, superimposed over the face.

In the culinary arts, the tendency to stuff food with a little sample of itself follows closely the unconscious pattern I have called, for lack of a better name, prototopia. Little Caesar's "Pepperoni and Cheese Stuffed Crust Pizza" (a pepperoni pizza whose crust is filled with pepperoni pizza) is perhaps the newest and most notoriously unhealthy example of gastronomic prototopia to this date. As for architecture, our obsession with loft spaces

constitutes a typically *prototopian* kind of obsession. Loft apartments are normally converted industrial spaces that retain the chic industrial looks (structural beams, exposed bricks and ductwork, etc.) of the factories and workplaces they once were. This non-division between work and leisure, between the public and the private, is a proto space that is already there before the place has been turned into living quarters. The middle-class buyer of a loft apartment is very aware that their home is the result of a displacement. Many jobs need to have been outsourced to a company in Bangladesh or Taiwan for a loft apartment to be possible in the first place. However, there is always the hope that the visibility of its structural traits—those high ceilings, exposed bricks, and wooden beams—can work either as a homage to the long-lost Fordist era or even as an implicit criticism of its post-Fordist replacement.

The visibility of societal structure within the structure of the apartment (the place within a place, a *mise en abyme*) is paramount to the internal design of the prototopian device. At the end of the day, prototopia is nothing but a delayed effect of what Louis Althusser called "uneven development" (Althusser et al. 2016, pp. 300–19). In late-capitalist social formations, criticism tends to be a part of the object criticized, insofar as the neoliberal edifice relies on an uneven relationship between the economic instance and the ideological one. It is, indeed, possible to distinguish two different stages in the development of late capitalism: first neoliberalism introduced itself as a revolutionary culture against standardized economics, and then it demanded a revolutionary counterculture to pair with—and antagonize—an already revolutionized economic order. Of course, the critique that "loft aesthetics" wields is to some extent valid: exposed brick walls and unpolished floors render the construction process visible against, say, upscale high-rise apartments, which proudly showcase the result and conceal the work invested in attaining it. However, the homage becomes caught up in a vicious circle when it becomes a homage to itself. In trying to restore the factory aura of the building, the loft-dweller acknowledges that the displacement has already taken place, indulging in the nostalgic vindication of a *present* time. We have here what Fredric Jameson baptized as "nostalgia for the present", which can also be exemplified by contemporary dramas filmed in black-and-white, or by pastiches that leech off dead styles to glorify the inevitable "here and now" (Jameson 1989, pp. 517–37).

Whether we consider it a mere cash-grab, an imaginary facelift, or a genuine exercise on nostalgia, the revamp of futuristic sci-fi classics (e.g., *Battlestar Galactica* or *Dr. Who*) that I mentioned above is but another example of that ubiquitous "here and now". Down-to-earth extraterrestrials who dress and talk like us, trivial technology, your everyday global pandemic, etc. Neo-futurist movies imagine a future that is already inhabited by the present; they are not utopian (they do not envision a better place), they are not dystopian (they do not portray a utopia gone wrong); they are prototopian, which means, in practical terms, they feature a world *that is just the way it is*. Such tautological status, however, cannot be treated as something empirically given and immediately available to perception. No object would be available to perception without the persistent layering process that organizes the surface level of capitalist realism, the imaginary level where surfaces interlace and overlap to bestow objects its very specific objectivity. It is precisely the constant overlapping of temporalities that draws my attention to time-travel fiction now. As utopias remain unimaginable and dystopias begin to wane, time-travel narratives are rising as the new bulwark of capitalist realism.

What kind of science fiction do we need today? The answer should be clear at this point: the "prototopian variety", since that is the form capitalist realism has chosen and is therefore its default home court, the proper battleground where the ideological war may be waged. However, not every time-travel narrative is going to be the same. Capitalist realism sets the framework, the silent mandate, and the hidden façade, but we can always make choices within that previously established framework. I want to examine these choices by comparing two recent Spanish time-travel fictional works: the massive TV hit *El ministerio del tiempo* (*The Ministry of Time*) directed by Javier and Pablo Olivares (2015–2020) (Olivares and Olivares 2015) and the coeval, direct-to-Netflix movie *Durante la tormenta* by Oriol

Paulo, translated into English as *Mirage* (2018). The difference between these works is, I claim, the way they relate to what Franco Berardi calls *futurability*: the ability of a present event to develop into its own future. Futurability concerns, in other words, not the ability of the present to jump from this moment to another entirely different one we identify as future, but the ability of the present to be the future itself.

### 3. Time-Traveling to the Present (and beyond)

The future is not what it used to be. It is certainly not a destination anymore. Time travel movies are, in this regard, among the most symptomatic cultural artifacts that are available to the average consumer of very symptomatic cultural artifacts today. If we look at the myriad time-travel movies that have been released over the last decades, we cannot but confirm a surprisingly conspicuous pattern. Time travel today means traveling to the past to prevent the present from suffering an alteration, whether we are talking about the rise of the machines (*The Terminator*, 1984), a gruesome crime (*Cronocrímenes* by Nacho Vigalondo, 2007), or a cute date gone awry (*About Time*, 2013). The same applies to dozens of movies such as *The Time Traveler's Wife* (2009), *Looper* (2012), *Edge of Tomorrow* (2014), or *The Tomorrow War* (2021), which have in common being clueless about what tomorrow might look like. A classic such as *Back to the Future* may appear to be a movie about people traveling to the future, but appearances are deceiving: the main character, Marty McFly, travels to the past to facilitate his parents' marriage, and only then goes back to the future, which is indeed the present. No one seems to be able to travel to the future anymore. If Kraftwerk's retrofuturism or Sun Ra's Afrofuturism dreamed of a past that paved the way into the future in the 1970s, contemporary science fiction can only seem to aspire to a future that eerily looks like the past.[8]

*El ministerio del tiempo* is no exception to this rule of rules. A government agency patrols the doors of time so that no intruder from another of the many eras to which the doors lead can change history for their benefit. Its motto could be summarized by the title of the first episode, "el tiempo es el que es", which was wrongly translated into English for the international distribution of the show as "time is what it is" (el tiempo es *lo que* es). The title does not refer to abstract, Newtonian time; it refers to concrete, tangible, material time, also known as history. The present history of Spain must not be changed, and by present history the Olivares brothers mean, among other things, the political deadlock inherited from the "times" of the Spanish Transition.[9] After all, the argument goes, there would have not been any progress if the high officials and military elites of the Francoist regime had not been onboard from moment one to co-pilot the democratic project. This is the conservative magic of the series: regardless of what its position towards historical events (say, the Spanish Transition) may be, the show will adhere to the same conservative principle: to keep time as it is.[10] If the deal had been something such as "not too much democracy and not too much dictatorship: just a healthy balance between the two", this deal would have been as good back in 1978 as it is now, because both 'then' and 'now' belong to the same "timeline" that needs to be constantly upheld.[11]

Hence, the Ministry of Time is the police of history, or history as Jacques Rancière's *police*: their agents partake in a state apparatus entrusted to guarantee the continuity of this ideological middle ground against any possible intrusion of the past (i.e., "revisionism") that might beg to modify the present.[12] Therefore, maybe, on second thought, the wrong translation is right on point after all. There is no history: there is only time. *El ministerio del tiempo* incurs in a double naturalization, the naturalization of political economy as historical teleology, and the naturalization of history as cold, hard, merciless time. Time is both "objective" and "inevitable" in this series. In the kingdom of rust deployed by the Olivares brothers, the positivistic approach to history is not at odds with a good old Baroque *contemptus mundi*. Perhaps that is why so many prominent figures from the so-called Spanish Golden Age make their stellar appearance along the way: Cervantes, Lope de Vega, Velázquez, or the Count-Duke of Olivares himself.

A very detailed close reading of several episodes of the show could very well support this interpretation, which I regard as almost self-evident, but it would not add much to the mix. Therefore, I will expand in another direction. In my view, the reason why *El ministerio del tiempo* is a neo-conservative experiment (and a very successful one) is not that it openly roots for the preservation of the current status quo, which is, of course, the essence of any kind of conservatism. The reason why *El ministerio del tiempo* is a conservative work is its lack of *potency* to recreate the different futures that are already inscribed into the present state of things. We tend to imagine conservatism as a reservoir of inherently conservative values (for instance, the "Golden Age" values that identify a certain idea of Spain), but this is a blatantly redundant notion. Conservatism is not the cause, but the effect. It might not even have any values for all that matters. Once deprived of the attributes that conceal this void, conservatism may be stripped down to a simple gesture: the one that ties possibility and actuality together, as a result of which a very particular passion arises.

Franco Berardi calls this passion *impotence*, defined as the inability to see that the present is impregnated with a plethora of possibilities (or, in Deleuzian jargon, "singularities") that power has rendered invisible: "the individual organism is cleared of any mark of singularity and transformed into a smooth surface, free of roughness" (Berardi 2019, p. 55). To be sure, the smooth surface or tabula rasa whose constituents are permanently aligned with each other throws us back into the universe of prototopia. That is why, in *El ministerio del tiempo*, the different doors in the present give access to different moments in the past, but every door in the past can only give access to the same moment in the present. Singular moments are not singular: they belong to the body of time, just as every store, place, and person in a mall belongs to the architecture—the original texture, the edifice without levels where everything is leveled—of the mall. However, Berardi's most interesting contribution to the issue of singularity concerns the notion of perception itself. The lexicon of distortion and invisibility has always been associated with that of ideology, be it in its strictly Hegelian definition as false or alienated consciousness or in the more sophisticated conception of ideology as a language that is always already structuring reality. This still holds true today, or truer than ever in a general sense, but Berardi takes two steps further in an interesting direction.

On the one hand, he makes the case for a distinction between the determination of potency and impotence. The difference is probably, from a Lacanian point of view, less significant than Berardi would like to admit, but his argument nevertheless stands: one thing is to say that ideology works like a language that determines life, and another thing is to state that life is that language, without any kind of mediation or determination required. That is what Berardi calls, following Jean Baudrillard, semio-capitalism.[13] On the other hand, Berardi emphasizes the importance of time as a battlefield. After neoliberalism knocks down the fourth wall of waged labor, which is the imaginary wall separating production from non-productive activities, time emerges as the new target of emancipation. Having renounced to the Fordist aspiration to full employment, which was made possible, among other factors, by a markedly sexual and racial division of labor, the fight for rightful employment is substituted by the fight for liberating time from the hold of capital. Huge marginal profits resulting from accumulated labor and technological progress (the same progress that had killed the jobs in the first place) should suffice to alleviate the burden of waged work for a large segment of the population. However, what about collective time? What about humanity's "work schedule"? The move, as far as capitalism is concerned, does not differ much from the management of individual time. Capitalism cannot survive without cherishing the idea of the future, yet it sees itself as the threshold beyond which no future is thinkable. The narrative solution is the "back to the future" plotline I mentioned above: the need to stage a hypothetical situation in which progress is possible only inasmuch as it leads to the present (conceived of as future).

Once again, this is nicely exemplified in *El ministerio del tiempo*. As observed before, there is a time door for every period in the history of Spain, but no door leads to the future. The explanation for this comes as early as in the first episode of the series when newcomer

Julián asks Salvador Martí (the chief officer) if it is possible to travel to the future. Martí answers negatively, adding enigmatically that "this is the time that is" (Quirós 2015, 21:46). Although Martí is not flat-out lying (he means that one cannot travel beyond 2015), this is just not true within the fictional coordinates of the series itself. If fellow time agents Amelia Folch (a nineteenth-century intellectual) or Alonso de Entrerríos (a seventeenth-century soldier) had been present during the course of the revelation, they could have confirmed that it is indeed very possible to travel to the future. Otherwise, how did they join the Ministry of Time in the present? Similarly, in that first episode, a French general from Napoleon's army has traveled from 1808 to 2015 to find out the denouement of the Peninsular war. Therefore, *El ministerio del tiempo* does not suggest that traveling to the future is impossible; it only emphasizes that the only future to which one may travel is the present. Asked about this obvious plot hole, Javier Olivares settles the question by arguing that "every series or every novel has its own rules", and that, for *El Ministerio del tiempo*, "the temporal limit is the present" (Quirós 2015, 29:00). Naturally, the real answer to this question is that these rules are not Olivares' rules, but the rules of capitalist realism, or the rules that cannot be bent within its limits. In the era of impotence, the first ones to be regarded as impotent should be the creators themselves.

How can we, then, *trespass* the boundaries of the present? The question is rather inconsequential because it is constantly being answered in numerous works of fiction. Take, for instance, *Durante la tormenta* by Oriol Paulo (2018). In 1989, during the fall of the Berlin Wall, a young boy named Nico is recording himself playing Cindy Lauper's "Time After Time" on his electric guitar. A violent electric storm does not muffle the noise coming from outside. Nico hears someone screaming and sees his neighbors having an argument through the window. Driven by curiosity, he goes into the house only to find out that his neighbor, Ángel Prieto, has murdered his wife, Hilda. Nico tries to run away and is eventually hit by a car. He dies instantly. However, in 2014, a married couple who have moved to Nico's former house along with their daughter find the old TV set Nico was using to record himself the night of his death, during the storm. Vera turns it on and sees Nico playing guitar. An identical storm has begun, and it has somehow connected both time frames through a wormhole. Vera, who knows the tragic story about the child's death 30 years before, warns him not to go out to his neighbor's house. She has saved his life, but in doing so, she has altered the chain of events that configures the present. Now, she is not married, and she does not have a daughter, although she remembers her husband and her daughter too well. What happened? Nico fell in love with Vera as a child, found her, and prevented her from meeting her husband, David. In this new version of the present, Nico and Vera are married. The rest of the movie accompanies Vera's rather conventional quest for the old TV set to reconnect with young Nico again and tell him not to obsess about her. She will succeed and end up reuniting with her family the day after the storm.

The movie is nothing otherworldly. Set in a very affluent suburb that looks like anything but Spain (or Catalonia, for that matter), it is infested with cheap thriller twists, cardboard characters with aspirational names, and gross inconsistencies. To mention just one: how does Vera get to meet Nico in the Vallpineda train stop, outside of Barcelona, if she only moved to that residential area after marrying David? What is she doing there if she is single?[14] Nevertheless, and despite its many flaws, *Durante la tormenta* does something time-travel movies do not dare to do that often: it raises the issue of futurability. If this is a time-travel movie, the question we should ask ourselves is: has Nico traveled to the future? Technically, he has not. Compliant with the restrictions neoliberalism and its "no future" mantra impose over time travel, he just saw a future woman on TV and patiently waited for her at the station.[15] However, the present in which he lives has been *interrupted* by the future, filled-in with the possibility of an event to come that is, from the very moment in which the storm connects the two time-periods, not exterior but interior to it. As Jacques Derrida would say, quoting Shakespeare, in *Durante la tormenta*, "time is out of joint" (Derrida 1995, pp. 14–38), a feeling—a déjà vu, maybe—that must resonate strongly with the viewer of Spanish cinema in recent times.

During the last two decades (and maybe more, but I am choosing Guillermo del Toro's *El espinazo del diablo* as a point of departure), we have beheld the boom of "hauntology" movies in Spain; movies about a present that was never present to itself because it was haunted by the past (the Spanish civil war, the unresolved issues of the Spanish transition, etc.). Though narratively satisfying and fundamentally correct in their assessment of history, these movies come off as predictable and dull, not to say politically ineffective. A quick search on the internet will return dozens of testimonies criticizing their bold allegorical layout, their Manichean and overall biased treatment of the past, etc. The reenactment of trauma seems to elicit the opposite reaction these movies are going for: that of a loop effect that helps the past catch up with the present without touching it. They bring about not the exorcism of memory, but its possession: a tendency to turn history into memory, into a fetishized fragment of the past that shuts out the way into the future.[16] With all the due respect for and sympathy towards this cinematic discourse, one must wonder if the critique of the Spanish Transition is not the ultimate and most refined example of the very same Culture of Transition it seeks to target and destroy. Read against the backdrop of *El ministerio del tiempo*'s massive success, *Durante la tormenta* is a refreshing take on the worn-out theme of time travel for reasons that have to do with the *mobilization* of memory: rather than another movie about a present haunted by the past, we are greeted with a movie about a present haunted by the future.

These things notwithstanding, *Durante la tormenta* does not transport us to a nowhere land beyond the ambush of capitalist realism. It concedes that the storm has not concluded; that whatever we do must be done while the storm is still underway. If today is the terminal limit beyond which it is impossible to make progress, the year 1989 (the year Francis Fukuyama proclaimed the end of history) is the terminal limit against which the loop of capitalist realism throws us away once and over again. We simply bounce back. However, the movie suggests that we can return to 1989 and start anew now that we have the experience of the future. The present is just but one of the multiple versions of a present that is not present to itself, not the best possible present, and certainly not the original one. It is, in fact, a forgery: David cheats on Vera, Vera is not the *true* Vera or the Vera she is supposed to be (the neurosurgeon she could have been), Nico never became a police inspector, a killer is still on the loose, etc. The "revisionist" reading, on the other hand, the present intervened by the future, the present presented as variability and impregnated with a multitude of conflicting singularities, is not only possible but also necessary. Unlike *El ministerio del tiempo*, *Durante la tormenta* champions a constant reworking of the present; this second chance given to Nico will allow us to see things the way they really are: David is a compulsive liar, Ángel is a psychopath, and Vera will eventually meet Nico, who is (we are forced to believe this) the love of her life. Things are what they are only if time is not what it is.

A harmless cinematic example will never make much of a difference, let alone one that does not even try to make it. There is no such thing as a narrative solution that may be applied to everyday life like a mathematical formula; there are only narrative queries to which reality responds in a certain way. Fiction can provide, best-case scenario, a lead that may be followed to help us map out the possible. Oriol Paulo's film offers a futuristic plot without a future, a time-travel experience without the trip, a flick about alternative realities without the dialectical leap of faith. Is this not what a sensible leftist agenda should look like today? That is, *theoretically*, how impotence turns into potency: by activating the dormant intensities that bridge potentiality and actuality together, by retelling the same story—the last remaining story—as if it had already happened in the future. The opposite gesture is a plausible narrative choice: we may persist in transiting the road that the left has followed for decades, which is no other than to assume, in a traditionally futuristic vein, that pointing at those magic islands of pure political truth will suffice to make them appear on the horizon. However, the story is going to sound old, as science-fiction repeatedly keeps proving, if not directly retro or démodé. Some stories simply cannot be told anymore. We do not tell others something is possible, such as the anti-globalization movements did

in the late 1990s to no avail, or like the Barack Obama campaign did in the late 2000s with equally disappointing results. We tell them it is already done, and we proceed to describe the inevitable.

**Funding:** This research received no external funding.

**Institutional Review Board Statement:** Not applicable.

**Informed Consent Statement:** Not applicable.

**Data Availability Statement:** Not applicable.

**Conflicts of Interest:** The author declares no conflict of interest.

## Notes

1   Of course, a Hegelian would rush to contend that negativity in its relation to itself is identity, as Hegel himself noted in his *First Principle* (Hegel 1869, p. 9). However, is this not precisely the problem? The anti-anti-lion could never be anything more than a lion, no matter how deep into the rabbit (or anti-rabbit) hole negativity is willing to take us.

2   I am referring, of course, to Fukuyama's infamous claim that late capitalism constitutes the last stage of the Enlightened enterprise, and therefore "the end of history". See Fukuyama (1989).

3   To delve into the Russian formalist notion of estrangement as applied to contemporary science fiction, see Spiegel (2008).

4   Benson's book compiles hundreds of predictions made by scientists and experts about what the future would hold. The predictions were published in the *Popular Mechanics* magazine between 1903 and 1969. Their forecasts were either ruefully funny or uncanny; sometimes prescient and sometimes absurdly utopian, but most times both.

5   This is not to say that the Chinese social credit system and the situation portrayed in *Black Mirror* are identical, among other things (critics of this comparison may be symptomatically forgetting this little detail) because the situation portrayed in *Black Mirror* is fictional. Therefore, we know very little about it. We usually know very little about things that do not exist, or at least not enough to make claims about the extent to which comparisons with existing systems or entities are empirically accurate. What this claim—my claim, now—means is that the similarities between these two terms are strong enough to draw a reasonable comparison, and that this comparison has inevitably led to questioning the American credit system in daily life conversations as well as in serious research. *That* is a question of fact. See, for instance, Rettinger (2021, p. 27).

6   For an account of the found footage genre in contemporary Spain, see Hardcastle (2017, pp. 108–23).

7   I am referring, naturally, to the *Anti-Oedipus* (Deleuze and Guattari 2000, pp. 9–16).

8   A brilliant analysis of the "retro aesthetics" that has dominated popular culture over the last few years can be found in Reynolds (2011, pp. 3–55).

9   Episode 12 of the third season, entitled "Contratiempos", makes this position abundantly clear. For a discussion of *El ministerio del tiempo* as "historical memory"—in the vein of Andreas Huyssen's notion of "past presents"—see Rueda Laffond y Coronado Ruiz (Rueda Laffond and Ruiz 2016, pp. 94–96).

10   Symptomatically, Andrew Niccol's *In Time* (2011), a movie that explores a near future in which time and not money is the standard currency, names the police "timekeepers".

11   See Cascajosa Virino (2020, pp. 38–52).

12   To understand the politics/police dichotomy in recent political theory, see Rancière (1999, pp. 21–42).

13   For Berardi, semio-capitalism is to the production of psychical stimulation what industrial capitalism was to the production of goods. Production may still be at the base, but value dwells simply elsewhere, as it is better described as the result of cognitive production. A much more eloquent explanation is found in Bray's chapter "Unproductive Worth" (Bray 2020, pp. 68–104).

14   For those who are not familiar with the area, real estate properties in the real Vallpineda are selling for up to two million euros, according to the website Idealista.com (accessed on 4 April 2022). Paulo opts for a very idealistic (not to say fraudulent or inauthentic) canvas of Spain.

15   As it is very well known, Margaret Thatcher's political opponents nicknamed her TINA to honor the dozens of times she had pronounced, or implied, the sentence "there is no alternative". That is the ineluctable paradox of a discourse—the neoliberal discourse of freedom—to which there cannot be any feasible free alternative.

16   I am in partial agreement here with Ángel Loureiro's reading here: "the debate is not truly about knowledge of the past but, in its best and most well-meaning instances, it is about new ways of viewing history itself, about a new sense of history as grievance" (Loureiro 2008, p. 227).

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
