# Peer review of "Futurism without a Future: Thoughts on The Ministry of Time and Mirage (2015–2018)"

_humanities, doi:10.3390/h11020058_

Round 1

Reviewer 1 Report

This article is fascinating.  It is not a typical scholarly article; it has much broader philosophical goals, and it redefines the contemporary sense of time and the genre of science fiction in a persuasive manner.  It does not examine TV and film texts for the sake merely of interpreting them, but rather to support these new ideas, which fit well into the context of contemporary philosophy (Zizek et al.)  I really liked the way the article veered skillfully away from the expected generic rules of the scholarly manuscript and became more universally appealing and more accessible.  At times there seemed to be jumps in the argument that took the reader somewhere else, but they were not overly jarring.  I am guessing the writer is in large part a philosopher/cultural historian with broad interests.  The Works Cited list shows the depth and breadth of the writer's knowledge and dispels any doubts that may remain about the scholarliness of the article.

Reviewer 2 Report

Overall, I thought that this was an engaging article and I enjoyed reading it. I do, however, have some comments for consideration and suggestions for improvement.

  • In order to meet the reader’s expectations in line with the chosen title, I felt that there should have been more analysis of the series (El ministerio del tiempo) and film (Durante la tormenta), and that Berardi’s concept of ‘futurability’ should have been foregrounded/ discussed earlier. It is not until page 8 that the analysis that I was expecting really begins. I also felt that the article would have benefited from greater engagement with the secondary sources on the series/film - especially on El ministerio del tiempo. (I was surprised that there was no mention of Cascajosa Virino’s work, for example). There could have been more commentary on the visual/ cinematographic effects in El ministerio/ Durante la tormenta too.
  • Some of the comments on realism are, in my view, debatable. Is it definitely the case that the novels of e.g. Galdós and Alas ‘represented lives’, for instance? (See page 5) Are their works, in fact, social commentary conveyed through the vehicle of fiction?
  • I wondered whether ‘post-apocalyptic’ (page 5) would have benefited from a definition (or at least some reference to secondary sources). Ditto with regard to ‘place’ (page 7).
  • It might be useful to include a reference to The Time Traveller’s Wife in the list of films mentioned on page 8?
  • With regard to the comments on the mistranslation of ‘el tiempo es el que es’ (page 8), I think that it would be useful to include the correct translation.
  • As regards the plot of Durante la tormenta and the author’s comment that this is ‘a movie about a present haunted by the future’ (page 11), is it equally possible that it is left to viewers to imagine what Vera’s future following the film’s ending might be (and is this necessarily ‘haunting’)? (Is it possible, for example, that the seeds of marital discord have been sewn and that she will leave David to pursue a life with Nico [and Gloria]?)
  • (Incidentally, regarding the author’s comments on the setting of Durante la tormenta on page 10, I wondered whether it is deliberately quasi-Americanized to appeal to a more universal audience?)
  • Finally, this may be a minor point, but I was somewhat puzzled as to why the author had chosen to put the translated titles in the main title, but the original Spanish titles in the abstract? (I wondered whether it might be best to include both the Spanish and English versions in the main title to ensure that the article is picked up by search engines if potential readers are looking for e.g. secondary sources on El ministerio del tiempo.)

I had some doubts about the style at times. It is generally engaging but a little too conversational (in my view) on some occasions. Some instances where I felt that the text would benefit from rephrasing include:

  • Page 1 line 29 - ‘First, a case of unintentional comedy.’ Change to e.g. ‘We will begin with a case of unintentional comedy.’
  • Page 4 line 159 - ‘(think of Soylent Green […].’ Change to ‘(such as…’)?
  • Page 4 line 162 - ‘Looking at it, […].’ Omit?
  • Page 4 line 197 - ‘let us just state the obvious for now […]’.
  • Page 5 line 209 – ‘to keep slowly dragging the discussion to the Spanish arena […].’
  • Page 5 line 230 – ‘One last example.’
  • Page 5 line 240-41 – ‘His is, no doubt about it, a laudable effort.’
  • Page 5 line 252 – ‘I will point out something for now […].’
  • Page 5 line 256 - ‘Take as an example, if you wish, […].’
  • Page 6 line 276 – ‘willy-nilly’
  • Page 6 line 280 – ‘Anyone who is not a total dummy, and I sincerely doubt that whoever is reading this article can qualify as a dummy at all, […].’
  • Page 6 line 297 – ‘let us remember this […].’ Omit?
  • Page 7 line 359 – ‘I want to examine […]’
  • Page 10 line 503 – should read ‘prevented her from meeting her husband’

Punctuation/ typographical errors

  • Page 3 line 146 – There should not be a full stop but a comma after “Ciudadanos”
  • Page 5 line 223 – should be ‘i.e.’
  • Page 11 line 577 –‘Barak’ should read ‘Barack
  • Is there a redundant inverted comma after ‘Verso’ in the Berardi entry in the list of references?
  • There is a gap that needs to be removed in the Rettinger entry (in the list of references). The main words of the article should also be in capital letters.
  • The main words of the article should also be capitalized in the Rueda Laffond etc. entry in the list of references.
